# Addition of a Blend Based on Zinc Chloride and Lignans of Magnolia in the Diet of Broilers to Substitute for a Conventional Antibiotic: Effects on Intestinal Health, Meat Quality, and Performance

**DOI:** 10.3390/ani12233249

**Published:** 2022-11-23

**Authors:** Gabriela M. Galli, João V. Strapazzon, Maiara S. Marchiori, Vitor L. Molosse, Guilherme L. Deolindo, Mateus Fracasso, Priscila M. Copetti, Vera M. Morsch, Matheus D. Baldissera, Ricardo E. Mendes, Marcel M. Boiago, Aleksandro S. da Silva

**Affiliations:** 1Animal Science Graduate Program, Universidade do Estado de Santa Catarina (UDESC), Beloni Trombeta Zanini, n 680-E, Santo Antonio 89815-630, SC, Brazil; 2Department of Animal Science, Universidade do Estado de Santa Catarina (UDESC), Beloni Trombeta Zanini, n 680-E, Santo Antonio 89815-630, SC, Brazil; 3Graduate Program in Toxicological Biochemistry, Universidade Federal de Santa Maria (UFSM), Av. Roraima, n 1000, Cidade Universitária, Camobi 97105-900, RS, Brazil; 4Graduate Program in Pharmacology, Universidade Federal de Santa Maria (UFSM), Av. Roraima, n 1000, Cidade Universitária, Camobi 97105-900, RS, Brazil; 5Laboratory of Veterinary Pathology, Instituto Federal Catarinense, Rodovia SC 283—km 17, Concordia 89703-720, SC, Brazil

**Keywords:** antimicrobial, intestinal health, plant components, productive efficiency

## Abstract

**Simple Summary:**

The additive blend based on zinc chloride and lignans from magnolia, mainly at the dosage of 50 mg/kg, can replace conventional antimicrobials as growth enhancers. This blend improved intestinal health and meat quality, two beneficial factors for the poultry industry. The improvement in meat quality resulting from the intake of the blend based on zinc chloride and lignans obtained from magnolia is desirable to consumers.

**Abstract:**

This study aimed to determine whether adding a blend based on zinc chloride and lignans from magnolia to the diet of broilers could replace conventional performance enhancers. For this study, 360 chickens were divided into four groups, with six repetitions per group (*n* = 15), as follows: CN, without promoter; GPC, control, 50 mg/kg of enramycin growth promoter; T-50, additive blend at a dose of 50 g/ton; and T-100, additive blend at a dose of 100 g/ton. Chickens fed with the additive blend at 50 g/ton showed a production efficiency index equal to that in the GPC group (*p* < 0.05). At 42 days, the lowest total bacterial count (TBC) was found in the T-100 group, followed by that in the GPC group (*p* < 0.001). For *E. coli*, the lowest count was observed in the T-100 group, followed by that in the CP and T-50 groups (*p* < 0.001). Higher villus/crypt ratios were observed in birds belonging to the T-100 and T-50 groups than in the GPC and NC groups (*p* < 0.001). Greater water retention was found in the T-50 group than in NC and T-100 groups (*p* < 0.048). The lowest water loss during cooking was also noted in the T-50 group (*p* < 0.033). We concluded that adding the antimicrobial blend, primarily at 50 g/ton, maintains the efficiency of the index of production and improves the intestinal health and meat quality of the birds.

## 1. Introduction

Worldwide, countries are adopting rules designed to reduce the use of antibiotics as growth promoters in poultry. Restrictions have also been imposed on the use of antibiotics as growth promoters in animal feed. The European Union banned the use of antimicrobials in 2006 [1]. Countries such as the USA, Canada, Mexico, Japan, China, and India have limited the use of antimicrobials as promoters in animal feed [2]. In Brazil, the second-largest producer and the largest exporter of chicken meat, there is also a partial prohibition of some of these agents [3,4]. There is growing pressure to adopt growth-promoting antimicrobials due to the possibility of cross-resistance of pathogenic bacterial strains between humans and animals.

Lignans are polyphenols that are found in the morphological parts of most plants. In particular, the vegetable lignans isolariciresinol, secoisolariciresinol, diglucoside, lariciresinol, and matairesinol [5] are of great interest. When ingested by animals, lignans are metabolized by the intestinal microbiota into enterolignans [6]. These polyphenols are being studied as possible alternative additives for their medicinal properties [7]. Lignans are classified as secondary metabolic plants, and they serve as protective agents. These were used for medicinal purposes in Asian countries 1000 years ago due to their anti-tumor, hepatoprotective, antifungal, sedative, antioxidant [7], and anti-arthritic properties, in addition to their ability to decrease the gene expression of inflammatory enzymes [8].

Magnolia is a plant widely used in human medicine; however, few studies have explored its effects on farm animals. Magnolia and its purified components have shown anticancer effects, as well as helped to treat various neuronal, inflammatory, cardiovascular, and gastrointestinal diseases [9,10]. Furthermore, magnolia can improve the zootechnical performance of broilers and reduce intestinal lesions caused by *Eimeria* spp. [11]. Park et al. [12] reported that magnolia extract could increase amino acid metabolites, such as lysine, and thus maintain intestinal homeostasis. However, there is still a significant gap in the literature regarding the effects of magnolia on broiler chickens. Zinc chloride was added the magnolia formulation due to its properties involved in synthesizing cholesterol, protein, and fats, as well as in supporting the immune and antioxidant systems. We hypothesize that combining these ingredients creates a synergistic effect, enhancing animal performance.

The objective of the present study was to determine whether the addition of a blend based on zinc chloride and the lignans of magnolia bark could replace conventional performance enhancers; we also measured the effects of this additive on zootechnical performance, microbial count, energy metabolism, serum biochemistry, antioxidants, and broiler meat quality.

## 2. Materials and Methods

### 2.1. Products

The tested product was purchased from Feedis (ROIMax^®^, Salto, São Paulo, Brazil). It is a blend based on zinc chloride and lignans used to support the digestive tract of poultry and swine. According to the technical data sheet, the product is composed of zinc chloride and lignans of magnolia bark, as well as silicon dioxide used as the vehicle. Magnolia is the primary ingredient, and the zinc in the blend formula stimulates anti-inflammatory and antioxidant effects, improving membrane integrity, reducing lesions, and consequently, improving nutrient absorption.

### 2.2. Animals, Accommodation, and Feed

The 360 one-day-old male chicks (46 ± 0.2 g) of the Cobb 500 strain were acquired from a commercial hatchery in Chapecó-SC, Brazil. Upon arrival at the experimental shed, the birds were immediately weighed and arbitrarily distributed in a completely randomized design. They were allocated into four groups, with six repetitions per group and 15 chickens per repetition (Figure 1). The birds were kept in 1.2 × 2 m pens on shavings beds and were exposed to a light program following the poultry lineage manual. The experiment took place in a negative-pressure aviary. The *Comitê de Ética do Uso de Animais na Pesquisa* (CEUA) of the Universidade do Estado de Santa Catarina approved the project under protocol number 7175220620.

The basal feed was formulated based on crushed corn and soybean meal (Table 1), according to the nutritional requirements for broilers described in the Brazilian Tables for Poultry and Pigs [13]. All groups received the same basal diet. The only changes were the addition of antibiotics and ROIMax. Therefore, the treatments were named CN, negative control without a promoter (only a basal diet); GPC, growth promoter control (a basal diet containing 50 mg/kg of enramycin); T-50, a basal diet supplemented with the additive blend at a dose of 50 g/ton; and T-100, a basal diet supplemented with the additive blend at a dose of 100 g/ton. The experimental period was 42 days, during which the experimental rations were supplied throughout the production phase (initial, growth, and termination phase), and water was offered ad libitum.

### 2.3. Performance

The chickens were weighed on days 1, 21, 35, and 42 of the experiment using a digital scale (grams). The feed intake (CR) (g/bird/day) was calculated as the difference between the feed provided at the beginning and the weight of the leftovers at the end of each period. The feed conversion (FC) was calculated by the total amount of feed ingested divided by the live weight of the birds. At the end of the production cycle, the productive efficiency index (IEP) was calculated according to the following formula [14]:$$\mathrm{IEP}=\frac{\mathrm{Body}\;\mathrm{weight}\;\times \;\mathrm{Viability}}{\mathrm{Age}\;\mathrm{at}\;\mathrm{slaughter}\;\times \;\mathrm{Feed}\;\mathrm{Conversion}}$$

### 2.4. Sample Collection

At day 42 of the experiment, blood was collected from 12 chickens per treatment, two per repetition. The chickens were manually contained, and using a 1 mL syringe, blood was collected from the ulnar vein. Subsequently, this blood was placed in tubes, without anticoagulants, to obtain the serum. This blood was centrifuged at 3500 rpm for 10 min, and the serum was separated, collected, and frozen (−20 °C) for biochemical analysis. At 42 days, six chickens per treatment were slaughtered: a broiler of average weight (at 42 days) was selected for slaughter from each box. Intestinal (jejunum) fragments were used from these animals for histology, oxidative and antioxidant status measurements, and energy metabolism enzyme evaluation. The pectoral muscle was collected for meat quality analysis and kept refrigerated until the analysis occurred 5 h after slaughter.

### 2.5. Histopathology

The jejunum samples were kept in bottles with a 10% formaldehyde solution. Fragments measuring 1 cm of thickness were collected, fixed in 10% buffered formalin (Merck KGaA^®^, Boston, MA, USA, catalog number R03379-86) for 48 h, and dehydrated in increasing concentrations (70–80–90–100–100%) of ethyl-alcohol solutions (Sigma-Aldrich^®^, St. Louis, MO, USA, catalog number R8382-1GA) for a duration of 1 h in each concentration. The samples were then embedded in paraffin wax (78 °C per 3 h) and cut into sections ranging in thickness from 3 to 4 μm. According to the method of Cardiff et al., the tissue sections were stained with hematoxylin and eosin (H&E) [15]. Under a light microscope, the morphological structure of the intestinal portions collected was evaluated. The villus length and crypt depth were measured according to the methodology described by Caruso and Demonte [16]. Histological images of the slides were captured using a digital micro-camera (Electronic Eyepiece Camera Video, Shenzhen, Guangdong, China) coupled to a trinocular biological microscope (model TNB-41T-PL, OPTON, Beijing, China) and a specific program for capturing histological images (Images J). More details of the methodology used to measure villus length and crypt depth were described by Galli et al. [17].

### 2.6. Stool Microbiology

Poultry litter samples were collected on days 21 and 42 at five locations in the box. The samples were homogenized and analyzed as a pool. This material was refrigerated and processed in the laboratory 2 h after collection. We used 1 g of each excreta sample, weighed aseptically, diluted in 9 mL of buffered peptide water in a sterile test tube, and homogenized in a vortex shaker. This gave rise to 10^1^ dilutions, from which we generated dilutions up to 10^6^, per the methodology described by Galli et al. [17]. We then inoculated 1 mL of the 10^6^ dilutions of each sample on 3M ™ Petrifilm^®^ ™ plates (Labtec, Londrina, Paraná, Brazil) for TBC and *E. coli* counts; 100 μL were inoculated on count agar plates (according to the manufacturer’s instructions), followed by incubation for 24 h and 48 h, respectively, at 37 °C. The results were expressed as colony-forming units per g (CFU/g).

### 2.7. Oxidant/Antioxidant Status

Oxidative and antioxidant biomarkers were measured in small intestine homogenates. The tissue was weighed and subsequently homogenized in saline solution (1 v/9 v proportion), followed by centrifugation (8500× *g* for 10 min), and the supernatant was collected and stored in microtubes under freezing (−20 °C) until analysis.

To determine the levels of reactive oxygen species (ROS) in the intestine, the methodology described by Halliwell and Gutteridge [18] was followed. The levels of non-homogeneous jejunum lipid peroxidation were determined as levels of antibodies reactive to thiobarbituric acid (TBARS), measured by the absorbance of the red product at 532 nm, according to the method described by Ohkawa et al. [19], and expressed as nmol MDA/mg of protein.

GST activity was measured using spectrophotometry at 340 nm, according to the methods of Habig et al. [20]. The mixture contained jejunum homogenate supernatant as the test sample, 0.1 M potassium phosphate buffer (pH 7.4), 100 mM GSH, and 100 mM CDNB, used as substrate. Enzyme activity was expressed as µmol/CDNB/mg protein. Protein thiols (PSH) were determined by the method described by Sedlak and Lindsay [21], a technique that uses DTNB (5,5-dithiobis-2-nitrobenzoic acid; Sigma^®^, São Paulo, Brazil). Protein thiols were measured by the sediment formed by the precipitated protein in which the material was resuspended, using a homogenization buffer to determine the PSH content. The absorbance readings (405 nm) were performed using a spectrofluorometer (Biotek, Synergy HT, Winooski, VT, USA).

### 2.8. Energetic Metabolism Enzyme

The intestinal homogenate supernatant was also used to measure energy metabolism enzymes. The intestinal protein level content was evaluated using the Coomassie blue G dye method [22], with serum bovine albumin as the standard. Creatine kinase (CK) activity was assayed in the reaction mixture containing the following final concentrations: 65 mM Tris-HCl buffer, pH 7.5, 7 mM PCr, 9 mM MgSO_4_, and 20 μL of the sample. After 10 min of pre-incubation at 37 °C, the reaction was started by the addition of 0.3 μmol of ADP and stopped after 10 min by the addition of 1 μmol of ρ-hydroxymercuribenzoic acid. The creatine level was estimated according to the colorimetric method described by Hughes [23]. The color was developed by adding 0.1 mL of 2% α-naphthol and 0.1 mL of 0.05% diacetyl in a final volume of 1 mL, and the results were recorded at 540 nm after 20 min. The results were expressed as U/L.

Pyruvate kinase (PK) activity was assayed essentially as described by Leong et al. [24]. The incubation medium consisted of 0.1 M Tris/HCl buffer, pH 7.5, 10 mM MgCl_2_, 0.16 mM NADH, 75 mM KCl, 5.0 mM ADP, 7 U L-lactate dehydrogenase, 0.1% (*v*/*v*) Triton X-100, and 20 μL of the sample, in a final volume of 500 μL. After 10 min pre-incubation at 37 °C, the reaction was started by adding 1 mM PEP. The results were expressed as nmol of pyruvate formed per min per mg of protein.

### 2.9. Serum Biochemical Analysis

The concentrations of total protein, albumin, triglycerides, cholesterol, uric acid, glucose, alanine aminotransferase (ALT), and aspartate aminotransferase (AST) were determined using commercial analytical kits (Analisa^®^, Palmitos, Santa Catarina, Brazil) and a semi-automatic biochemical analyzer (Bioplus 2000^®^, Barueri, Brazil). The serum globulin levels were calculated using the mathematical formula globulin = total proteins − albumin.

### 2.10. Meat Quality Analysis

The chest muscle was removed and maintained under refrigeration for five hours. After this period, we obtained the meat, and the pH was measured using an electrode, as well as the value of L (brightness), an (intensity of red), and b (intensity of yellow) in the samples, using a Minolta Chrome Meter. Water-holding capacity and weight loss from cooking (WLC) were measured, as described by Hamm [25] and Honikel [26], respectively. The samples resulting from the WLC analysis were used to determine the shear force (SF) using cuts of 1.5 cm in width, with muscle fibers oriented in the direction perpendicular to the blade of the equipment (Texture Analyzer TA-XT2i, Surrey, UK), coupled to a device (Warner-Bratzler, Kansas, United States) that provides the ideal force necessary to cut the sample [27]. The HR results were expressed as kgf/cm.

### 2.11. Statistical Analysis

All variables were subjected to the normality test (Shapiro–Wilk). Then, the data were subjected to analysis of variance and Tukey’s test, in which a significant difference between treatments was considered when *p* < 0.05 using software R, version number 20220930 (SAS package, São Paulo, Brazil).

## 3. Results

### 3.1. Performance

In the initial phase (1–21 days), there were no differences between groups in regards to body weight, feed consumption, or FC (*p* > 0.05). From days 1–35, there were no differences related to body weight or FC; and from days 1–42, there were no differences in regards to body weight (*p* > 0.05). From days 1–35, the GPC group showed a lower feed consumption than did the NC group (*p* < 0.042). From days 1–42, a higher feed consumption was observed in the T-100 group than in the GPC and T-50 groups (*p* < 0.001). Meanwhile, the lowest FC in the same period was observed in the GPC and T-50 groups, compared to those found in the NC and T-100 groups (*p* < 0.001; Table 2). A higher productive efficiency index (PEI) was found in the chickens from the GPC and T-50 groups than in the chickens from the NC group (*p* < 0.001; Table 3).

### 3.2. Histology

The highest villus height was observed in the T-100 group compared to the chickens in the GPC and NC groups (*p* < 0.001). Greater crypt depths were found in the T-100 group than in the CP (*p* < 0.001). Higher villus/crypt ratios were observed in T-100 and 50 groups than in the GPC and NC groups, while the GPC group exhibited ratios superior to those in the NC group (*p* < 0.001; Table 4). No lesions or changes were observed in the intestines (Figure 2).

### 3.3. Bacterial Effect

At 21 days, lower TBC levels were observed in samples from chickens belonging to the T-50 and T-100 groups than those in the NC group, and the GPC group exhibited lower counts than those in the NC group (*p* < 0.001). At 42 days, the lowest TBC count was obtained in the birds from the T-100 group, followed by those in the GPC group, compared to the NC group (*p* < 0.001). The lowest *E. coli* count at 21 days was observed in the GPC, T-50, and T-100 groups, and at 42 days, the lowest count was observed in the T-100 group, followed by the GPC and T-50 groups as compared to that in the NC group (*p* < 0.001; Table 5).

### 3.4. Oxidant/Antioxidant Status

In the intestine, higher ROS levels were found in the T-50 and T-100 chickens than in those from the GPC and NC chickens (*p* < 0.001) (Table 6). Higher levels of TBARS were found in the GPC group, followed by those in the NC group, and the lowest level was observed in the T-100 group (*p* < 0.001) (Table 6). Lower GST activity was observed for birds in the GPC, T-50, and T-100 groups than in the NC group (*p* < 0.046). There were no differences for the TSH variable (*p* > 0.05) (Table 6).

### 3.5. Energetic Metabolism Enzyme

Higher levels of CK in the intestine were found in the PC and T-100 groups than in the NC group (*p* < 0.010); the same effect was observed for the enzyme PK (*p* < 0.001). There were no differences between groups regarding the TP variable (*p* > 0.05; Table 6).

### 3.6. Serum Biochemistry

Lower levels of serum cholesterol were found in the T-50 and T-100 groups than in the GPC and NC groups (*p* < 0.011); the same effect was found for uric acid (*p* < 0.030). For ALT, the lowest concentration was found in the T-100 group, followed by those found in the GPC group, and the highest level was found in the NC group (*p* > 0.05). Lower concentrations of AST were found in the T-50 and T-100 groups than in the GPC and NC groups (*p* < 0.001). There were no differences for the biochemical variables of total protein, albumin, globulin, triglyceride, or glucose (*p* > 0.05; Table 7).

### 3.7. Meat Quality

Higher luminosity was observed in the T-100 than in the T-50 and PC groups (*p* < 0.037). Higher water retention was observed in the T-50 group than in the NC and T-100 groups (*p* < 0.048). The lowest water loss by cooking was found in the T-50 group compared to that in the NC group (*p* < 0.033). A lower SF was found in the T-100 group than in the GPC group (*p* < 0.003). No differences were observed for the pH or the intensities of red and yellow variables (*p* > 0.05; Table 8).

## 4. Discussion

The inclusion of a growth promoter blend based on zinc chloride and lignans obtained from magnolia at 50 mg/kg in the diet of broilers improved the chickens’ performance compared to that of the NC group. This compound can replace antibiotics without the loss of performance. The poultry chain uses the productive efficiency index to generate value for the integrated producers of these industries [28]. This assessment considers the integration between body weight, viability, age of the birds, and FC. This index shows that the inclusion of 50 mg/kg of the additive provided performance equal to that of the birds that received the performance enhancer. In this context, a blend based on zinc chloride and magnolia bark provided a better villus/crypt ratio and may have stimulated enzyme secretions, thus improving the digestion and absorption of nutrients.

The excellent performance resulting from the blend consumed by the broilers is related to the improvement in intestinal health. Chen et al. [29] reported that 200 mg/kg magnolol supplementation increased occluding junction protein expression and reduced nitric oxide production, increasing intestinal villi in laying hens. This can improve the intestinal barrier due to the decrease in intestinal permeability, leading to a reduction in the entry of pathogens and toxins into the intestinal lumen, which impairs nutrient absorption and consequently, reduces the zootechnical performance of the birds.

Some microorganisms, toxins, and anti-nutritional factors can affect the structure of the villi and intestinal crypts. The increase in the height of the villi occurs with advancing age; however, it can be stimulated by the increased activity of the digestive enzymes [30]. This fact may have occurred in the present study because the vegetable components may have influenced the activity of some enzymes, consequently improving the digestion and absorption of nutrients. However, the improved performance is usually attributed to the control of pathogenic microorganisms leading to a lower severity of intestinal infections and less competition for nutrients from the host. However, this did not occur in the T-100 group, suggesting that the ideal dose to enhance growth performance is less than 100 g/ton. Furthermore, a greater crypt depth was found in this group, which suggests higher protein turnover [31] in response to desquamation or inflammation induced by pathogens, toxins, or anti-nutritional factors; this represents an energy expenditure and an associated increase in the size of the intestinal villi. Therefore, our findings may represent a compensatory mechanism for higher protein turnover, increasing the absorption area.

In the present study, using a blend based on zinc chloride and lignans from magnolia, among other components, decreased the microbial counts of *E. coli* and the TBCs. Marino et al. [32] tested various plant extracts, particularly active components such as polyphenols, coumarins, lignans, steroids, and terpenes. These authors observed a bacteriostatic effect of these extracts against *S. aureus*, as they decreased the formation of biofilm and the initial adhesion of cells. Tufano et al. [33] showed that the antimicrobial activity of lignans might be due to the presence of hydroxyl and furan rings in the chemical structure; however, the exact mechanism remains uncertain.

Cui et al. [34] reported that the antibacterial effect of the magnolia extract is due to the inactivation of enzymes related to the intracellular metabolism of bacteria. Chen et al. [35] observed that magnolol increased the richness and diversity of the microbiota of 21-day-old broiler chickens challenged with *S. pullorum*. Therefore, this fact may explain the decrease in the TBC and *E. coli* found in the present study because, by increasing beneficial bacteria, there is a decrease in pathogenic bacteria through competition between the two.

The enzymes CK and PK catalyze cell phosphotransfer reactions that mediate communication between the cytosol and the nucleus and are, therefore, essential for cell homeostasis [36]. These enzymes catalyze the exchange of nucleotides, facilitating movement between the places of generation using ATP (adenosine triphosphate) [37]. The enzyme pyruvate kinase (PK) is involved in the metabolism of glycolysis [38], which converts phosphoenolpyruvate into pyruvate. Thus, the increase in PK activity may have stimulated the generation of ATP in the intestine. Baldissera et al. [39] reported the relationship between an increase in PK and an increase in the production of pyruvate and ATP, which can increase energy transfer and distribution. CK is a protein that acts on cells that have high energy demand, with the function of transferring a phosphoryl group from ATP to creatine to produce adenosine diphosphate (ADP) and phosphocreatine (PCr) [40]. Thus, the increase in CK activity may mean an increase in the source and transport of energy. In addition, it is essential to reduce pathogenic bacteria, as the integrity of the epithelial cells depends on energy. Therefore, mitochondrial bioenergetic dysfunction of the enterocytes occurs in infections [30]. In this context, an increase in the PK and CK enzymes related to the phosphotransfer network was observed, constituting a beneficial effect of the additives.

In typical physiological situations, cells generate and release quantities of ROS, hydrogen, and sulfur that act as important signaling molecules and contribute to immune functions such as programmed cell death [41]. Sánchez-Villamil et al. [42] found that innate immune cells, such as macrophages, produce free radicals to control pathogens. In this context, it is known that there is a regulation of the oxidant and antioxidant status by the body. This mechanism may explain the increase in ROS in the groups that received the additive; possibly, this increase occurred to combat pathogenic microorganisms, a fact observed by the lower *E. coli* and fecal coliforms counts in the birds.

TBARS levels are used as an indicator of lipid peroxidation. Hu et al. [43] reported a positive correlation between TBARS and *Prevotella*. Thus, there may have been an increase in this genus in the GPC group, which may be the cause of the increase in TBARS. Chabane et al. [44] supplied methomyl to rats and observed that it exerts peroxidative damage on the intestinal cell membranes. This same phenomenon may have occurred due to the use of enramycin. The body possesses defense mechanisms, including enzymatic and non-enzymatic antioxidants, to reduce the damage caused by free radicals. Among these enzymatic antioxidants, the GST enzyme is critical. It has the auxiliary function of protecting against toxic substances and metabolites [45]. The decrease in GST activity in the T-50, T-100, and GPC groups may be due to the use of GSH as a substrate, since we observed an increase in ROS or TBARS in these groups. Several intermediate metabolites are generated in the detoxification processes, possibly explaining the reduction in GST activity [46].

The reduction in serum cholesterol in birds from the T-50 and T-100 groups may be associated with an increase in *Lactobacillus* spp. in the intestines [47] due to the reduction of pathogenic bacteria such as *E. coli*. For *Lactobacillus* spp., these can bind cholesterol to cell surfaces and convert intestinal cholesterol to coprostanol [48]. In addition, these herbal additives can inhibit the activity of the enzyme 3-hydroxy-3-methyl glutaryl-CoA reductase, a key enzyme in the synthesis of cholesterol; thus, their inhibition reduces levels of serum cholesterol [49].

Uric acid is the end product of nitrogen metabolism, in addition to being an endogenous antioxidant. Uric acid indicates the efficiency of amino acid utilization in broilers [50]. Therefore, increases in uric acid are related to increased protein catabolism [51]. Therefore, we believe that the T-50 and T-100 groups were more efficient in protein utilization due to lower endogenous protein turnover and, thus, reduced endogenous nitrogen and ammonia losses. The enzymes AST and ALT indicate liver damage. Thus, the increase in the activity of these transaminases indicates damage to hepatocytes, as was observed in the CN and GPC groups. Lignans may have had a hepaprotective and antioxidant effect.

Lignans can act by decreasing the concentration of cytoplasmic Ca^2+^, thereby inactivating caspase 2. Furthermore, the non-transference of excess cytoplasmic Ca^2+^ to the mitochondria does not promote the release of cytochrome C, and the activation of caspase 9 and 12, which are responsible for the activation of caspase 3, induces hepatocyte apoptosis [52]. Therefore, inactivating this pathway decreased hepatocyte apoptosis. Finally, the antioxidant effect of lignans is due to the presence of hydroxyls in their molecules. Magnolol and honokiol found in magnolia exhibit antioxidant activity, as they participate in the reduction of ONOO^-^ in 1O_2_; they donate the hydroxyl groups present in their molecules to free radicals [53], which could explain the decrease in liver enzymes. These specific mechanisms may have occurred in this context, which explains the reduction in liver enzymes in birds that received the additive.

Differences in meat quality between studies can be attributed to several factors: pH, sarcomere length, ionic strength, osmotic pressure, and the development of rigor mortis, all of which alter cellular and extracellular components [54]. Greater luminosity was observed in the T-100 group; therefore, a possible explanation would be that the neutral poles lose their ability to interact with the water in the sarcoplasm; this phenomenon causes a lowered capacity to retain water during cooking, causing greater cooking losses. In this way, the lower amount of solute present in the sarcoplasm causes the water to leave the intracellular medium for the extracellular medium. During this transport, there is an increase in humidity at the meat’s surface, which causes greater luminosity due to the greater reflection of the incident light [55].

The water-holding capacity directly influences the color and tenderness of the meat. The increase in the water retention capacity observed in the T-50 group means that there was an increase in the water content present in the muscle, increasing the succulence of the meat [54], which is also related to lower protein denaturation [56]. The lowest loss of water during cooking was observed in the T-50 group, which can be attributed to the lower loss of moisture and fat during cooking, a fact related to the ability of the protein matrix to retain water and prevent the migration of lipids. Increasing the capacity to retain water and reducing water loss by cooking are desirable factors for both the poultry industry and poultry consumers. They represent economic losses as they cause changes in the product’s composition (vitamins and amino acids) and nutritional value [56].

The lower SF in the T-100 group can be explained by the larger myofibers, as they have a lower glycolytic potential [57]. Therefore, the increase in the diameter of the fibers is related to an increase in the tenderness of the meat. In addition, muscle denaturation reduces the proteins myosin and actin, resulting in softer meat [58]. Another possible explanation would be the higher content of intermuscular fat [59] in the meat of the birds belonging to the T-100 group, a factor also related to tenderness [60]. Lin et al. [61] reported that supplementing 200 and 300 mg/kg of *Magnolia officinalis* extract decreased water due to cooking loss and increased water retention capacity in duck meat due to its antioxidant effects, retarding lipid and protein oxidation.

## 5. Conclusions

The additive blend based on zinc chloride and lignans from magnolia, mainly at 50 mg/kg, can replace conventional antimicrobials as growth enhancers. Antimicrobial action was identified, which suggests that this product fulfills its purpose. In addition, this additive improved intestinal health and meat quality, two beneficial factors for the poultry industry. Finally, the improvement in meat quality is desired by consumers.

## Figures and Tables

**Figure 1 animals-12-03249-f001:**
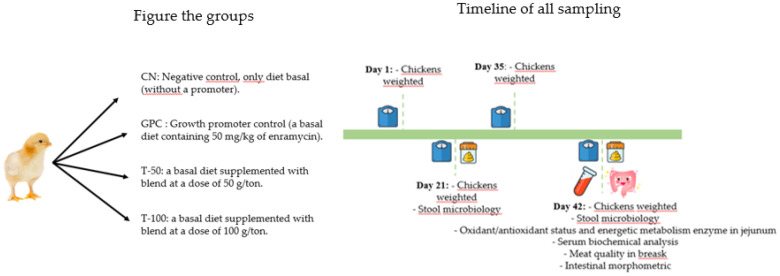
Schematic showing the groups and the timeline of all sampling.

**Figure 2 animals-12-03249-f002:**
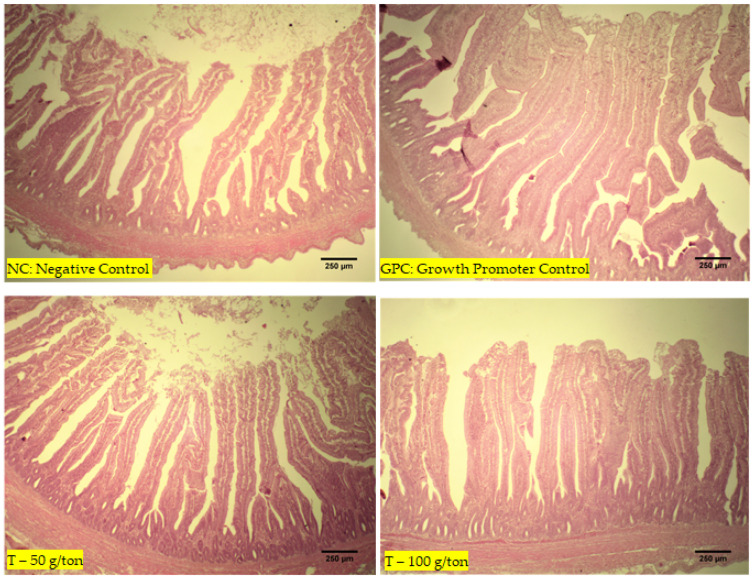
Images of the intestine (jejunum) of chickens from the four treatment groups at 42 days of age, all without alterations or lesions.

**Table 1 animals-12-03249-t001:** Ingredients in the basal diet used for all experimental groups.

Ingredients (kg/ton)	Age (Days)		
	1–21	22–35	36–42
Corn	551.51	580.05	621.35
Soybean meal	373.01	337.00	298.22
Soy oil	39.05	49.19	49.70
Dicalcium phosphate	12.71	13.00	112.25
Calcitic lime	11.42	9.28	8.00
Iodized salt	4.86	4.23	3.95
DL-Methionine—99%	2.91	2.82	2.50
L-Lysine—78%	2.03	1.95	2.63
L-Threonine—99%	0.50	0.48	0.40
Premix of vitamins and minerals ^1^	2.00	2.00	2.00
Calculated chemical composition	100	100	100
Digestible energy (kcal/kg)	3050	3150	3200
Crude protein (%)	21.20	19.80	18.40
Calcium (%)	0.84	0.76	0.66
Available phosphorus (%)	0.40	0.35	0.31
Digestible lysine (%)	1.22	1.13	1.06
Digestible methionine (%)	0.47	0.45	0.42
Digestible methionine + cysteine (%)	0.88	0.83	0.77
Digestible threonine (%)	0.79	0.73	0.69
Digestible tryptophan (%)	0.21	0.20	0.19
Sodium (%)	0.21	0.20	0.19
Zinc (ppm)	60.0	60.0	60.0

^1^ Minimal vitamin and mineral levels per kg of product: vitamin A (5,000,000 UI); vitamin D3 (1,000,000 IU); vitamin E (15,000 UI); vitamin K3 (1500 mg); vitamin B1 (1500 mg); vitamin B2 (3000 mg); vitamin B6 (2000 mg); vitamin B12 (7000 mcg); folic acid (500 mg); nicotinic acid (15 g); pantothenic acid (7000 mcg); choline (80 g); biotin (100 mg); copper (10 g); iron (50 g); iodine (1000 mg); manganese (80 g); selenium (300 mg); zinc (70 g); minimum humidity (20 g); maximum mineral matter (980 g). Growth promoter (enramycin, 10 mg/kg of feed); coccidiostatic agent (salinomycin, 64 mg/kg of feed).

**Table 2 animals-12-03249-t002:** Performance (mean and standard deviation) of the broilers fed an antimicrobial blend to replace growth promoters.

Day 1 to 21			
Treatment	Body weight gain (g) ^2^	Feed intake (g)	FC
NC: Negative Control	1007 (17)	1367 (46)	1.42 (0.06)
GPC: Growth Promoter Control	998 (35)	1398 (36)	1.47 (0.05)
T—50 g/ton	998 (38)	1386 (36)	1.45 (0.06)
T—100 g/ton	993 (51)	1381 (118)	1.45 (0.11)
*p*-value ^1^	0.736	0.687	0.254
Day 1 to 35			
Treatment	Body weight gain (g)	Feed intake (g)	FC
NC: Negative Control	2573 (173)	3987 ^a^ (140)	1.58 (0.07)
CPG: Growth Promoter Control	2528 (88)	3778 ^b^ (196)	1.53 (0.03)
T—50 g/ton	2526 (132)	3808 ^ab^ (139)	1.51 (0.07)
T—100 g/ton	2550 (152)	3994 ^ab^ (325)	1.59 (0.06)
*p*-value ^1^	0.941	0.042	0.057
Day 1 to 42			
Treatment	Body weight gain (g)	Feed intake (g)	FC
NC: Negative Control	3318 (101)	5336 ^ab^ (235)	1.63 ^a^ (0.05)
GPC: Growth Promoter Control	3389 (87)	5151 ^b^ (207)	1.54 ^b^ (0.04)
T—50 g/ton	3440 (37)	5195 ^b^ (153)	1.53 ^b^ (0.05)
T—100 g/ton	3484 (105)	5572 ^a^ (226)	1.62 ^a^ (0.06)
*p*-value ^1^	0.062	0.001	0.001

^1^ Different letters in the same column indicate significant differences between groups according to the Tukey test. ^2^ Initial weight of birds (day 1): NC—Negative Control (45.97 g), GPC—Growth Promoter Control (46.01 g), T—50 g/ton (46.04 g), and T—100 g/ton (45.98).

**Table 3 animals-12-03249-t003:** Production efficiency index values of broilers fed an antimicrobial blend to replace growth promoters.

Treatment	PEI
NC: Negative Control	478.3 ^b^
GPC: Growth Promoter Control	521.8 ^a^
T—50 g/ton	533.7 ^a^
T—100 g/ton	504.7 ^ab^
*p*-value	0.001

Different letters in the same column indicate significant differences between groups according to the Tukey test.

**Table 4 animals-12-03249-t004:** The villus, crypt, and villus/crypt ratio of the jejunum of broilers fed an antimicrobial blend to replace growth promoters.

Treatment	Villus (µm)	Crypt (µm)	Villus/Crypt (µm)
NC: Negative Control	1308 ^b^	252 ^ab^	5.18 ^c^
GPC: Growth Promoter Control	1265 ^b^	211 ^c^	5.97 ^b^
T—50 g/ton	1506 ^ab^	225 ^bc^	6.67 ^a^
T—100 g/ton	1936 ^a^	290 ^a^	6.66 ^a^
*p*-value	<0.001	<0.001	<0.001
CV (%)	8.87	4.79	3.78

Different letters in the same column indicate significant differences between groups according to the Tukey test.

**Table 5 animals-12-03249-t005:** Total bacterial count (TBC) and *Escherichia coli* (EC) in poultry litter of broilers fed an antimicrobial blend to replace growth promoters.

Treatment	TBC (×10^6^ CFU/g)	TBC (×10^6^ CFU/g)	*E. coli* (×10^6^ CFU/g)	*E. coli* (×10^6^ CFU/g)
	Day 21	Day 42	Day 21	Day 42
NC: Negative Control	116.6 ^a^	495 ^a^	25.9 ^a^	102 ^a^
GPC: Growth Promoter Control	89.3 ^b^	414 ^b^	14.2 ^b^	56.7 ^b^
T—50 g/ton	62.2 ^c^	432 ^ab^	13.7 ^bc^	63.1 ^b^
T—100 g/ton	58.8 ^c^	342 ^c^	9.74 ^c^	38.7 ^c^
*p*-value	<0.001	<0.001	<0.001	<0.001
CV (%)	20.8	26.3	10.6	12.7

Different letters in the same column indicate significant differences between groups according to the Tukey test.

**Table 6 animals-12-03249-t006:** Oxidant/antioxidant and energetic metabolism enzyme status in the homogenate of the jejunum of broilers fed an antimicrobial blend to replace growth promoters.

Treatment	ROS ^1^ (×10^3^)	TBARS ^1^	GST ^1^	TSH ^1^	TP ^1^	CK ^1^	PK ^1^
NC: Negative Control	25.7 ^b^	2.64 ^b^	5.99 ^a^	1.21	3.90	0.79 ^b^	0.31 ^b^
GPC: Growth Promoter Control	35.9 ^b^	4.47 ^a^	4.36 ^b^	2.06	4.35	1.27 ^a^	0.47 ^a^
T—50 g/ton	65.8 ^a^	2.17 ^bc^	4.73 ^b^	1.54	4.22	1.06 ^ab^	0.41 ^ab^
T—100 g/ton	59.2 ^a^	1.43 ^c^	4.64 ^b^	1.70	3.78	1.15 ^a^	0.49 ^a^
*p*-value ^2^	0.001	0.001	0.046	0.084	0.102	0.010	0.001
CV (%)	16.7	4.10	5.74	10.9	9.32	3.86	2.07

^1^ Note: ROS—reactive oxygen species (U DCF/mg protein); TBARS—thiobarbituric acid reactive substances (mmol MDA/mg protein); GST—glutathione S-transferase (μmolCDNB/min/mg protein); TSH—protein thiols (µmol TSH/mg protein); TP—total protein (mg/L); CK—creatine kinase (nmol creatine formed/min/protein); PK—pyruvate kinase (nmol pyruvate formed/min/protein). ^2^ Different letters in the same column indicate significant differences between groups according to the Tukey test.

**Table 7 animals-12-03249-t007:** Biochemistry of the serum of broilers fed an antimicrobial blend to replace growth promoters.

Treatment	TP ^1^	ALB ^1^	GLO ^1^	TRI ^1^	CHO ^1^	UA ^1^	GLU ^1^	ALT ^1^	AST ^1^
NC: Control	4.18	1.98	2.20	157	187 ^a^	6.36 ^a^	168	4.20 ^a^	341 ^a^
GPC: Growth Promoter Control	4.46	1.96	2.50	159	193 ^a^	6.58 ^a^	223	1.80 ^b^	350 ^a^
T—50 g/ton	4.08	2.14	1.94	164	149 ^b^	5.32 ^b^	150	1.48 ^bc^	230 ^b^
T—100 g/ton	3.64	1.64	2.00	155	148 ^b^	5.18 ^b^	240	0.81 ^c^	229 ^b^
*p*-value ^2^	0.214	0.123	0.097	0.589	0.011	0.030	0.064	0.050	0.001
CV (%)	5.86	6.29	5.17	8.79	7.36	8.68	24.9	16.74	12.7

^1^ Note: TP—total protein (g/dL); ALB—albumin (g/dL); GLO—globulin (g/dL); TRI—triglyceride (mg/dL); CHO—cholesterol (mg/dL); UA—uric acid (mg/dL); GLU—glucose (mg/dL); ALT—alanine aminotransferase (U/L); AST—aspartate aminotransferase (U/L). ^2^ Different letters in the same column indicate significant differences between groups according to the Tukey test.

**Table 8 animals-12-03249-t008:** Chemical-physical variables in the meat of broilers fed an antimicrobial blend to replace growth promoters.

Treatment	Color L	Color A	Color B	pH	WRC ^1^	WLC ^1^	SF ^1^
NC: Control	52.8 ^ab^	–1.86	11.9	5.75	75.8 ^b^	24.6 ^a^	2391 ^ab^
GPC: Growth Promoter Control	50.0 ^b^	–1.56	11.4	5.93	76.6 ^ab^	21.2 ^ab^	2633 ^a^
T—50 g/ton	50.2 ^b^	–1.28	11.9	5.75	79.8 ^a^	19.8 ^b^	2306 ^ab^
T—100 g/ton	55.0 ^a^	–1.93	12.3	5.76	74.9 ^b^	22.1 ^ab^	2019 ^b^
*p*-value ^2^	0.037	0.814	0.473	0.127	0.048	0.033	0.003
CV (%)	2.39	2.86	1.96	4.74	3.79	3.94	6.87

^1^ Note: WRC—water retention capacity (%); WLC—water loss by cooking (%); SF—shear force (cm^2^). ^2^ Different letters in the same column indicate significant differences between groups according to the Tukey test.

## Data Availability

Raw data are held by the authors and may be available upon request.

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
