# Peer review of "Addition of a Blend Based on Zinc Chloride and Lignans of Magnolia in the Diet of Broilers to Substitute for a Conventional Antibiotic: Effects on Intestinal Health, Meat Quality, and Performance"

_animals, 2022, doi:10.3390/ani12233249_

Round 1

Reviewer 1 Report

Line 1: determine the type of article

Line 21-36: According the journal, the front matter should be made up of a simple summary of no more than 200 words and an abstract of about 200 words maximum, in a single paragraph and should follow the style of structured abstracts but without headings

https://www.mdpi.com/journal/animals/instructions

Introduction: must be expanded and made more linear, at the moment it is confusing

In this study authors want to try zinc chloride, silicon dioxide, and magnolia bark. In the introduction, authors introduce only magnolia but not zinc chloride and silicon dioxide. what is the connection with the lignans?

Line 78: add a table showing the composition

Line 81: add the authorization number of working with animals

Line 82: explain why only males were used and this strain

Line 84: add the weight

Line 85: describe the statistical analysis according to which authors decide this sampling

Line 136-137: are samples embedded in paraffin? Are samples cut at microtome and at what thickness? Add a reference for hematoxylin and eosin staining.

Line 214: add a scheme/figure showing the groups and the timeline of all sampling

In the section of materials and methods, add for all reagents used the number catalogue and the company

Line 242: add a figure showing histology results comparing all groups

Line 250: change table 4 with table 5

References: rewrite references following instructions provided by the journal

https://www.mdpi.com/journal/animals/instructions

Author Response

Comments and Suggestions for Authors

Line 1: determine the type of article

OK

Line 21-36: According the journal, the front matter should be made up of a simple summary of no more than 200 words and an abstract of about 200 words maximum, in a single paragraph and should follow the style of structured abstracts but without headings.  https://www.mdpi.com/journal/animals/instructions

Answer: ok. adjusted

Introduction: must be expanded and made more linear, at the moment it is confusing.In this study authors want to try zinc chloride, silicon dioxide, and magnolia bark. In the introduction, authors introduce only magnolia but not zinc chloride and silicon dioxide. what is the connection with the lignans?

Answer: Reviewer, the text has been proofread; and new information has been added to the introduction section. If you have another suggestion, let me know.

Line 78: add a table showing the composition.

Answer: I don't understand, table 1 already contains the composition of the diets. if you can clarify, thank you.

Line 81: add the authorization number of working with animals

  1. added

Line 82: explain why only males were used and this strain

Answer: Went uses only broilers males for standardization of the weight

Line 84: add the weight

  1. added

Line 85: describe the statistical analysis according to which authors decide this sampling

Answer: Our experimental shed has a maximum capacity of only 24 boxes. This sample n allows a post-hoc power between 87 and 92% when considering meat quality variables.

Line 136-137: are samples embedded in paraffin? Are samples cut at microtome and at what thickness? Add a reference for hematoxylin and eosin staining.

Answer: Yes. Added

Line 214: add a scheme/figure showing the groups and the timeline of all sampling

Answer: Yes. Added

In the section of materials and methods, add for all reagents used the number catalogue and the company

Answer: ok. added information

Line 242: add a figure showing histology results comparing all groups

Answer: ok. added in supplementary material (figure S1)

Line 250: change table 4 with table 5

Answer: Added

References: rewrite references following instructions provided by the journal

https://www.mdpi.com/journal/animals/instructions

  1. revised

Reviewer 2 Report

The manuscript entitled “Addition of a blend based on zinc chloride, silicon dioxide, and magnolia in the diet of broilers to substitute for a conventional antibiotic: effects on intestinal health, meat quality, and zoo-technical performance”by Galli et al, reported a study on the effect of addition of a blend based on zinc chloride, silicon dioxide, and magnolia replacing enramycin on growth performance, gut health and meat quality of chicken, the topic is meaningful for the reducing the use of antibiotics, but the manuscript is still in need for further improvement.

1, what is the meaning of zoo-technical performance in the title?

2, the mixed-use of birds, poultry, and chickens is imprecise, replaced the birds and poultry with chicks or chickens

3, the negative control is explained as without promoter, do you mean basal diet only?

4, CP also means crude protein, using CP as a group name is confusing.

5, L28-29, it is hard to understand, please rephrased this sentence.

6, L30, is this result based on your statistical analysis, what is the mean by reporting the P value?

7, L32-34,Please describe these indicators according to academic norms.

8, in the introduction section, why focus on the Lignans in a whole paragraph, and what do you want to support?

9, L67-68, there are quite plenty of researches on the effects of magnolia on chickens to my knowledge, so what is the novelty of this research?

10, L76-77, what do you mean by stating that ROIMax is a blend to support the digestive tract? And the detail information on ROIMax composition should be provided, especially, the zinc and lignans content in ROIMax.

11, table 1, the chemical composition is only calculated? Not measured value? And energy is based on gross energy or digestible energy or metabolism energy? Brute protein should be crude protein. Additionally, the zinc content should be provided.

12, L115, in the formula, IEP should be PEI? And where does the parameter viability come from? The formula should be referred to a more reliable source.

13, L118, you have 6 repetitions and collected blood from two chickens per repetition, why only ten blood samples are achieved? And the experiment was assigned to three production phases, why blood samples were collected only at end of the experiment at 42 days?

14, L119, what is the insulin syringe?

15, L123, how to choose the slaughtered chickens?

16, L124-125, and L159-172, it is confusing, what kind of samples you have harvested, and detailed information on the treatment of the samples is also lacking. 

17, L147, why not collected litter directly, and how is it performed by collecting litter samples at five different places in the box?

18, L153-154, what is the mean by stating six total bacterial counts? 100mL of what were inoculated?

19, L160, and L165, where does the muscle sample come from? From jejunum as your description in L124-125?

20, L174, what is the mean of energetic metabolism enzyme? are these enzymes measured in the digesta?

21, the model used in the statistical analysis is not provided.

22, table 2, the initial body weight should be provided. and with the days 1 to 21, do you mean the body weight at 21 days or body weight gain during days 1 to 21? 

23, table 4, 6, and 7, the unit should be provided.

24, L244, ambiguous information

25,English should be improved.

Author Response

Comments and Suggestions for Authors

The manuscript entitled “Addition of a blend based on zinc chloride, silicon dioxide, and magnolia in the diet of broilers to substitute for a conventional antibiotic: effects on intestinal health, meat quality, and zoo-technical performance”by Galli et al, reported a study on the effect of addition of a blend based on zinc chloride, silicon dioxide, and magnolia replacing enramycin on growth performance, gut health and meat quality of chicken, the topic is meaningful for the reducing the use of antibiotics, but the manuscript is still in need for further improvement.

1, what is the meaning of zoo-technical performance in the title?

Answer: Sorry, I removed the zoo-technical. I think an error occurred.

2, the mixed-use of birds, poultry, and chickens is imprecise, replaced the birds and poultry with chicks or chickens

Answer: Replaced birds and poultry by chicks or chickens

3, the negative control is explained as without promoter, do you mean basal diet only?

Answer: Yes, I added this information in the text.

4, CP also means crude protein, using CP as a group name is confusing.

Answer: Ok, I replaced by CPG: Growth Promoter Control

5, L28-29, it is hard to understand, please rephrased this sentence.

Answer: Ok, rewritten

6, L30, is this result based on your statistical analysis, what is the mean by reporting the P value?

Answer: ok. added P-value

7, L32-34,Please describe these indicators according to academic norms.

Answer: In these lines we have a brief conclusion. I don't understand what you suggest. If you have a writing suggestion, it is welcome.

8, in the introduction section, why focus on the Lignans in a whole paragraph, and what do you want to support?

Answer: Because the lignans make part the blend, observed in magnolia.

9, L67-68, there are quite plenty of researches on the effects of magnolia on chickens to my knowledge, so what is the novelty of this research?

Answer: However, few studies with lignans and in this case the association of both

10, L76-77, what do you mean by stating that ROIMax is a blend to support the digestive tract? And the detail information on ROIMax composition should be provided, especially, the zinc and lignans content in ROIMax.

Answer: information added. Magnolia (5.8%) is the main ingredient of this product considering the application and functions that lignans do in the intestine, with zinc (10.2%); and silica is a vehicle.

11, table 1, the chemical composition is only calculated? Not measured value? And energy is based on gross energy or digestible energy or metabolism energy? Brute protein should be crude protein. Additionally, the zinc content should be provided.

Answer: These are calculated values based on the formed diet and formulation program; we do not measure composition. We added the requested information.

12, L115, in the formula, IEP should be PEI? And where does the parameter viability come from? The formula should be referred to a more reliable source.

Answer: Ok, I fixed, should be IEP. Sorry, but the researcher Stringline is reliable

13, L118, you have 6 repetitions and collected blood from two chickens per repetition, why only ten blood samples are achieved? And the experiment was assigned to three production phases, why blood samples were collected only at end of the experiment at 42 days?

Answer: Great observation; it was a writing error (correct is 12 samples - adjusted). It would be interesting to collect blood on the 21st and 35th as well, but when we designed the project we thought of only one blood collection at the end of the experiment; same date birds were slaughtered and we also had tissue samples to analyze and compare with serum results. Collections during the production cycle interfere with the performance of the bird that had its blood collected; we have already verified this in previous studies by our group.

14, L119, what is the insulin syringe?

Answer: 1 ml syring, I changed the text.

15, L123, how to choose the slaughtered chickens?

Answer: Chickens within the average weight range of each pen were selected.

16, L124-125, and L159-172, it is confusing, what kind of samples you have harvested, and detailed information on the treatment of the samples is also lacking. 

Answer: ok. sentence was reformulated.

17, L147, why not collected litter directly, and how is it performed by collecting litter samples at five different places in the box?

Answer: Yes, that's right, collected from 5 different locations. I improved the wording to make it clearer.

18, L153-154, what is the mean by stating six total bacterial counts? 100mL of what were inoculated?

Answer: sentence was reformulated.

19, L160, and L165, where does the muscle sample come from? From jejunum as your description in L124-125?

Answer: Yes, I changed the text.

20, L174, what is the mean of energetic metabolism enzyme? are these enzymes measured in the digesta?

Answer: L122-124, these enzymes went measured in the jejunum.

21, the model used in the statistical analysis is not provided.

Answer: ok added

22, table 2, the initial body weight should be provided. and with the days 1 to 21, do you mean the body weight at 21 days or body weight gain during days 1 to 21? 

Answer: I want to refer to body weight gain during days 1 to 21. The initial weight of the chicks has been added to the footer of Table 2.

23, table 4, 6, and 7, the unit should be provided.

Answer: The unit is in the note, under the table.

Table 4 fixed.

24, L244, ambiguous information. Revised and adjusted

25,English should be improved. Sent for English proofreading after corrections, certificate attached.

Reviewer 3 Report

This study builds on and complements earlier findings about the effects of plant-derived polyphenol supplementation in broilers (Galli et al. 2020, 2021). In this study, authors determined the effect of a blend with magnolia as a source of lignans (50 g/ton or 100 g/ton), on productive efficiency, meat quality and oxidative stress, metabolic, microbiologic and indicators. The main results reached by the authors were that zinc chloride, silicon dioxide and magnolia blend supplementation (50 g/ton) improved productive performance, meat quality, and intestinal health.

This manuscript provides some valuable information about feed strategies to diminish the use of antibiotics in poultry. In general, the coverage of relevant literature is correct, the research question is clearly stated in Introduction section, and the selected methods are appropriated to answer it. However, I suggest to the authors some minor revisions. For this referee, the Discussion section should be improved since the information given is not clear in some points and results are not connected.

Comments

Would it be possible to provide some information about the possible influence on the results of zinc chloride and silicon dioxide?

For this referee, results obtained in T-100 are not clear. Please, clarify the sentence ‘It can be assumed that the highest dose of the additive caused dysbiosis’ (line 329’), since animals in T-100 showed the lowest TBC. Statement in line 438 should be also supported with obtained results.

Minor comments

Line 28, include abbreviation to total bacterial counts

Line 29, revise missed words ‘compared to CN and with…’.

Line 78, if possible, include more detailed composition.

Line 104, please, clarify if growth promoter was included in 4 treatments.

Line 115, change to IEP to PEI

Line 120-121, change ‘this material’ for samples or blood, it is confusing.

Line 130 and 133, change ºC to °C

Line 202, change ‘an’ to ‘a’

Line 238, change ‘then’ to ‘than’

Line 239 and 284. Please, include units in tables 4 and 7.

Line 298 Change ‘Cor’ to ‘color’ in L, a or b.

Line 369. Reference 41 is a review, try to use other more suitable to the text.

Author Response

Comments and Suggestions for Authors

This study builds on and complements earlier findings about the effects of plant-derived polyphenol supplementation in broilers (Galli et al. 2020, 2021). In this study, authors determined the effect of a blend with magnolia as a source of lignans (50 g/ton or 100 g/ton), on productive efficiency, meat quality and oxidative stress, metabolic, microbiologic and indicators. The main results reached by the authors were that zinc chloride, and magnolia blend supplementation (50 g/ton) improved productive performance, meat quality, and intestinal health.

This manuscript provides some valuable information about feed strategies to diminish the use of antibiotics in poultry. In general, the coverage of relevant literature is correct, the research question is clearly stated in Introduction section, and the selected methods are appropriated to answer it. However, I suggest to the authors some minor revisions. For this referee, the Discussion section should be improved since the information given is not clear in some points and results are not connected.

Answer: Dear reviewer, we made the adjustments you requested, as did 2 other reviewers; therefore, I inform you that there have been several changes in the paper. If any adjustments are pending, let me know what we'll do in the next round of review.

Comments

Would it be possible to provide some information about the possible influence on the results of zinc chloride and silicon dioxide?

Answer: Zinc yes, we believe that combined with magnolia potentiated weight gain. Silica was used only as a vehicle. We adjusted this in the manuscript to avoid confusion.

For this referee, results obtained in T-100 are not clear. Please, clarify the sentence ‘It can be assumed that the highest dose of the additive caused dysbiosis’ (line 329’), since animals in T-100 showed the lowest TBC. Statement in line 438 should be also supported with obtained results.

 Answer: sentences were reformulated.

Minor comments

Line 28, include abbreviation to total bacterial counts

Answer: Fixed.

Line 29, revise missed words ‘compared to CN and with…’.

Answer: Fixed.

Line 78, if possible, include more detailed composition.

Answer: was added more information

Line 104, please, clarify if growth promoter was included in 4 treatments.

Answer: Not, only GPC, growth promoter control (a basal diet containing 50 mg/kg of enramycin). Information in line 96.

Line 115, change to IEP to PEI

Answer: Fixed.

Line 120-121, change ‘this material’ for samples or blood, it is confusing.

Answer: Changed.

Line 130 and 133, change ºC to °C

Answer: Changed.

Line 202, change ‘an’ to ‘a’

Answer: Changed.

Line 238, change ‘then’ to ‘than’

Answer: Changed.

Line 239 and 284. Please, include units in tables 4 and 7.

Answer: The unit is in the note, under the table.

Table 4 fixed.

Line 298 Change ‘Cor’ to ‘color’ in L, a or b.

Answer: Of course, I changed the text.

Line 369. Reference 41 is a review, try to use other more suitable to the text.

Answer: We couldn’t find another reference. If you have a suggestion of another reference that covers this subject, let me know; that we will make the adjustments in sequence.

Round 2

Reviewer 2 Report

Table 3, PEI should be IEP? please check other parts in the manuscript.

Reviewer 3 Report

The manuscript has been improved. Thank you very much.